# Electromagnetic Sensing Techniques for Monitoring Atopic Dermatitis—Current Practices and Possible Advancements: A Review

**DOI:** 10.3390/s23083935

**Published:** 2023-04-12

**Authors:** Alexandar Todorov, Russel Torah, Mahmoud Wagih, Michael R. Ardern-Jones, Steve P. Beeby

**Affiliations:** 1Centre of Flexible Electronics and E-Textiles, School of Electronics and Computer Science, University of Southampton, Southampton SO17 1BJ, UK; rnt@ecs.soton.ac.uk; 2James Watt School of Engineering, University of Glasgow, Glasgow G12 8QQ, UK; mahmoud.wagih@glasgow.ac.uk; 3Clinical Experimental Sciences, Faculty of Medicine, University of Southampton, Southampton SO16 1DU, UK; m.aj@soton.ac.uk

**Keywords:** electromagnetic sensing, flexible wearable sensors, telemedical sensors, non-invasive monitoring, atopic dermatitis, transepidermal water loss, interdigitated capacitive sensor, neural networks, radio frequency reflectometry, near-infrared range spectroscopy

## Abstract

Atopic dermatitis (AD) is one of the most common skin disorders, affecting nearly one-fifth of children and adolescents worldwide, and currently, the only method of monitoring the condition is through an in-person visual examination by a clinician. This method of assessment poses an inherent risk of subjectivity and can be restrictive to patients who do not have access to or cannot visit hospitals. Advances in digital sensing technologies can serve as a foundation for the development of a new generation of e-health devices that provide accurate and empirical evaluation of the condition to patients worldwide. The goal of this review is to study the past, present, and future of AD monitoring. First, current medical practices such as biopsy, tape stripping and blood serum are discussed with their merits and demerits. Then, alternative digital methods of medical evaluation are highlighted with the focus on non-invasive monitoring using biomarkers of AD—TEWL, skin permittivity, elasticity, and pruritus. Finally, possible future technologies are showcased such as radio frequency reflectometry and optical spectroscopy along with a short discussion to provoke research into improving the current techniques and employing the new ones to develop an AD monitoring device, which could eventually facilitate medical diagnosis.

## 1. Introduction

In the last decade, there has been an increasing amount of innovation and interest in developing systems for personalized, at-home health care, assisted by advances in information and communication technologies also known as “e-health” or digital health [1,2]. Digital health encompasses a broad range of services such as providing remote specialist consultation and diagnosis, effective storage and management of patients’ clinical records and prescription of medicine via the Internet. Providing specialist medical care and advice through means of communication is especially beneficial for rural or remote areas or when the patient is unable to leave their homes.

One area in medicine that has not seen much advancement since the introduction of e-health is dermatology and the study of lesions and skin inflammation. Diseases such as atopic dermatitis or psoriasis have so far been monitored in person by a specialist dermatologist. In the case of atopic dermatitis, there is also no universally accepted treatment algorithm; thereby, prescribed treatments of the same patient may vary depending on the dermatologist or the institution the patient attends [3,4]. There have been attempts to introduce telemedical solutions based on machine learning and artificial intelligence (AI) to detect and diagnose melanoma [5,6,7,8] and psoriasis [9] by analyzing pictures of lesional (damaged) skin. This is a step toward achieving a complete quantitative diagnosis through e-health systems, but these detection systems have limitations with accuracy and require costly equipment. This highlights the need not only for communication-based solutions for remote healthcare but for portable home-based medical devices that can assist in the management and treatments of patients in vivo. These devices have to be compact, affordable and compatible with the daily home routine of the patient.

Atopic dermatitis (AD), also known as eczema, is a chronic, relapsing skin disorder, which is characterized by recurring skin inflammation and pruritus (itchy skin) which results in changes to the surface of the skin such as redness, swelling, thickening, scaling and vesiculation (small blisters) [10]. A lesion is a noticeable patch of abnormal skin on the human body, and atopic dermatitis (AD) patients can experience the symptoms of the disorder (primarily itch) even if they do not have any visible lesions—i.e., their skin is “non-lesional” [11]. AD is one of the most common skin disorders worldwide, affecting nearly 20% of infants and children and around 3% of adults. It has been proven to negatively affect the quality of life on the patients, hindering their performance of daily activities [12]. The key pathogenic factors of the disease are multiple and likely to vary in relative contribution between individuals. However, there is consensus that the primary abnormality in AD is a dysfunction of the epidermis. This is most prominent in the disturbance of the stratum corneum (SC) barrier [10]. The SC is the outermost layer of the skin and is the main barrier preventing against skin desiccation and challenges from the environment [13]. The SC is composed of laterally stacked corneocyte cells, which are flat keratin filaments inside an envelope of cross-linked protein [14,15]. The stability of the SC barrier, or epithelial/epidermal barrier as it is also known, depends on the structural integrity of the corneocyte cells. In healthy skin, the cells are densely packed, making the skin durable and flexible as well as less susceptible to tissue damage or diseases. In AD, deficiencies and perturbations in key proteins which regulate cellular adhesion complexes and the epidermal structure are altered or absent, resulting in breaches of the epidermal barrier [16,17]. This causes an increase in water evaporation from the skin, which is a process called transepidermal water loss (TEWL). The broken barrier also makes the skin more vulnerable to environmental triggers of skin inflammation including allergens and microbes, which can cause skin inflammation, redness and irritation when penetrating the barrier [18].

The characteristics of the epidermal barrier and the ‘skin hydration’ can be assayed using various methods such as external diffusion chambers, skin elasticity, electromagnetic and optical screening, all of which should provide a sufficient basis for monitoring the severity and extent of this disease [19]. Along with the measurement of water content, the vascular dilation associated with inflammation which can be seen as redness in white skin or darkening of darker skins, as well as surface scaling from keratinocyte disruption can be quantified. Machine learning through image recognition can be employed, much like the systems applied for melanoma diagnosis [5,6,7,8]. Alternatively, the impact of the condition on the lives of patients can also be measured. Since the lesions are often irritating and itchy, activity sensors can track how often patients scratch themselves, providing an indication of how severe their condition is [20,21,22].

The purpose of this review is to identify the current methods of evaluation and monitoring of AD, starting with the medical practices that include invasive techniques and then covering telemedical e-health approaches that allow for remote and non-invasive monitoring. This review also highlights other techniques that are applicable to the field of monitoring skin conditions and would advance the study of AD. Finally, a discussion on each method’s merits and demerits is presented along with a critical conclusion on the most prominent technique for AD monitoring.

## 2. Established Methods for Monitoring AD

### 2.1. Medical Examination and Tests

There are several diagnosis criteria established, which are used to determine the severity and treatment of AD and related conditions. Some of the most widely used include Hanifin and Rajka, the Millennium and the U.K. diagnostic criteria [4,23]. They are all based on a combination of visual inspections by a specialist with medical examinations such as blood serum, biopsy, and tape stripping. The lack of a unified empirical criteria means that diagnosis and distinction is sometimes subjective based on the opinion of the acting dermatologist. Due to the complex nature of the condition and the lack of understanding on what the underlying cause is, the medical tests cannot fully empirically prove the condition and distinguish it from similar ones such as psoriasis or contact dermatitis [24]. Moreover, these tests are invasive and can induce anxiety and stress in the patients, which are predominantly children [25]. This section serves as an overview of what medical examinations are currently conducted in hospitals and clinics—biopsy, blood serum extraction and tape stripping. Figure 1 presents an infographic of the different chemical biomarkers that can be obtained through blood serum, biopsy or tape stripping. The methods presented do not fall in line with the e-health directive, and their drawbacks highlight the need for further research into the area of novel AD monitoring solutions.

#### 2.1.1. Biopsy

Biopsy is a medical procedure in which a small sample of human tissue is sourced from a patient to be further examined to determine the presence of a specific disease. Obtaining a skin biopsy is a commonly used method in the diagnosis of skin cancer, as it provides the most insight into the extent of the condition [26]. Varying levels of specific molecules and compounds found within the biopsy may serve as evidence toward the presence of AD.

Wessler et al. present a study in which they linked increased levels of acetylcholine in 2 mm thick biopsies of AD patients [27]. Acetylcholine is a compound synthesized by epidermis (EP) cells (layer beneath the SC) that acts as a signaling molecule that regulates cell adhesion and contact. The increase in acetylcholine in AD patients was found to be 14-fold, which makes it a solid biomarker of the condition [27]. The study also showed that injecting more acetylcholine in the skin causes an itching sensation, similar to how AD pruritus feels. The researchers do not mention whether the increased levels apply only to lesional AD skin or to non-lesional skin as well.

Kägi et al. investigated the relationship between specific cytokine protein production and the emergence of atopic dermatitis [28]. They sampled biopsies of lesional and non-lesional skin of AD patients and compared the presence of these proteins. The result was a completed protein profile of both types of skin in AD patients, so that the condition can be discovered even before skin inflammation and lesions have appeared [28]. While this might be beneficial for early discovery, due to its invasive nature, it is a drastic solution to a “seemingly” healthy skin.

Salvador et al. states that biopsy is not a necessary method in diagnosing AD but can provide useful in distinguishing it among other conditions such as psoriasis, dermatitis herpetiformis, and cutaneous lymphoma [29]. Overall, biopsy is a stressful procedure that should resolve only to the most extreme conditions such as melanoma, and in the case of diagnosing AD, it is needless.

#### 2.1.2. Blood Serum—“Liquid Biopsy”

An alternative to the biopsy procedure is to extract serum from the blood (also known as “liquid biopsy”) and to measure the levels of Immunoglobulin E (IgE). IgE is an antibody that is produced by the immune system to combat allergens [30]. Thus, in the case of AD, a broken epidermal barrier would be penetrated more by outside influences and allergens, increasing the levels of IgE [11].

Kagi et al. and Liu et al. both researched this effect independently by extracting blood serum from the bodies of AD patients and concluded that high IgE levels are a biomarker for atopic dermatitis [28,31]. This is a better assessment than single point biopsy, but it still involves the need for invasive extraction that must be performed by a specialist and then assessed in the laboratory environment.

#### 2.1.3. Tape Stripping

Tape stripping is a procedure where samples of clear tape with an adhesive on one side are gently pressed against the skin and then removed to be observed under a microscope. The collected tissue is mainly lipids from the SC, including corneocyte cells, linking proteins and fatty acids [32]. Since AD is constituted of a dysfunction in the barrier between the SC and the epidermis, information about the lipids in the SC is useful in determining the severity of the condition.

Yamamoto et al. obtained tape strips of patients with AD at varying skin locations and discovered that AD patients have a decrease in the proportion of the ceramide protein, which is responsible for cell linkage [32]. Røpke et al. sampled strips from both lesional and non-lesional skin areas of AD patients and concluded that levels of cytokine and chemokine proteins differentiate between the two, establishing another biomarker for AD [33]. The tape strip sampling process is visible in Figure 2. Koppes et al. employed the same tests as Ropke et al. but included tests on healthy subjects to discover that the chemokine protein is a better biomarker than cytokine [33,34]. Although tape stripping is proven to be an accurate and useful tool for the non-invasive diagnosis of AD, it still needs to be performed in a laboratory environment, as sample contamination is a major risk. Moreover, there is not a standardized procedure in performing the stripping—Yamamoto et al. and Koppes et al. repeatedly press and release the sample by hand, whereas Ropke et al. use a constant pressure instrument for a single press [32,33,34]. Thus, tape stripping is not suitable for the purposes of e-health monitoring, as it is not possible to effectively reproduce these tests at a remote location or at home.

#### 2.1.4. Conclusions

Diagnosing and examining AD in medical practices is primarily completed through visual assessment by looking at the skin lesions after the symptoms of the condition have emerged. New research has revealed that there are other biomarkers of AD in the skin of patients—such as IgE, chemokine, ceramide, and cytokine proteins—that can provide an empirical estimation of the severity of the disease and in the case of cytokine can even predict the emergence of the condition without any apparent symptoms. These biomarkers can be obtained through the means of biopsy, blood serum extraction and skin tape stripping. These empirical examinations are much better than the conventional visual ones because they rely on measured data instead of a subjective approach. Unfortunately, all these tests can only be performed in a controlled laboratory environment and require complex equipment and a qualified lab technician to correctly assess the results. Furthermore, they are all invasive and in the case of biopsy very stressful for the patients. Thus, these methods cannot fall in line with the e-health directive of providing non-invasive and remote monitoring to patients around the world, and the need for a new generation of sensing technologies for skin condition diagnosis and monitoring becomes more apparent. Table 1 summarizes the current medical tests for AD diagnosis and highlights their drawbacks.

### 2.2. Telemedical and Non-Invasive Measurements

Telemedical and non-invasive techniques are highly sought after for monitoring patients and as presented in the introduction, in cases such as melanoma, they are gaining support to replace medical examinations [5]. They are also widely used in monitoring other medical issues and vital signs including heart failure [35], respiratory [36], heart rate variability [37], fall detection [38] and anxiety [39]. These examples do not require complex laboratory equipment or a controlled environment and in most cases can be performed in the comfort of the patient’s home [40]. Monitoring is performed typically using a standalone instrument or a wearable system, the latter of which can be used for continuous monitoring without obstructing the daily routine of the patient. The devices usually comprise a sensing/screening part read by a controller unit that exports data to a computer or network where the data can be easily accessed and interpreted by a doctor [41]. This section will highlight telemedical techniques reported in the literature for monitoring AD and comment on their accuracy in evaluating the severity and their reproducibility as a portable e-health device. These are grouped into subsection according to the specific biomarker that the techniques use to perform their measurements.

#### 2.2.1. Transepidermal Water Loss (TEWL)

TEWL is the rate of water evaporation from the skin (in g/m^2^/h), as water naturally permeates through the SC barrier. In healthy skin, this rate is minimal, and it varies based on the body site, but an average value of TEWL in healthy skin is between 5 and 15 g/m^2^/h [42]. In damaged skin such as the skin of an AD patient, the evaporation rate is higher, in the order of 15–25 g/m^2^/h, since the barrier function is compromised [43]. Subjects with AD have been reported to have increased values of TEWL, which results in their visibly drier skin [18]. Therefore, a measurement of this property would provide a sufficient basis for monitoring the severity and extent of this disease. TEWL can be empirically measured by positioning an enclosed chamber probe onto the skin [44]. The probe has a hydration sensor built in to detect the increase in humidity in the chamber, and the controller unit driving the sensor produces a value for TEWL [43].

Kim et al. used a commercial evaporimeter sensor (EP1, Servo Medical, Vallingby, Sweden) to evaluate severe lesions from five different skin sites of AD patients [45]. They discovered a significant correlation of severity with increased TEWL in all the five sites measured (postauricular (underside of ear), thigh, abdomen, forearm, popliteal fossa (underside of knee)). The results were compared to measurements of TEWL at the same sites in healthy subjects, confirming that a dysfunction in the skin barrier’s structure provokes higher TEWL values [45].

Werner et al. used the same commercial evaporimeter as Kim but included TEWL measurements of clinically healthy sites of skin on AD patients and compared them to the skin of subjects without the condition [46]. They discovered that TEWL values in AD patients are increased even in the non-lesional areas where there is no visual damage, inflammation, or redness present. This is an important distinction because it means that TEWL measurements can be used to predict the condition even before any symptoms have appeared. Given that current diagnostic procedures rely on the observation of already visible lesions and symptoms, TEWL measurements are much faster in predicting the outcome and can be used to apply appropriate treatment the soonest. Xiao et al. compared two different commercial evaporimeters (the handheld battery-powered VapoMeter^®^, Delfin, Kuopio, Finland and the wired pen-like AquaFlux^®^ Biox Systems Ltd., London, UK; see Figure 3) to explore the effect of applying extra hydration to the skin [47,48,49]. They detected varying levels of hydration through the application of suntan lotions with TEWL values decreasing after the application of skin lotion. This needs to be investigated further, but it affirms the claim that hand creams and moisturizers assist the barrier function of the skin; thereby, it is possible that constant application might decrease the severity of AD lesions.

Sim et al. constructed their own TEWL measurement device following the same principle of operation used by the commercial devices, but they manage to house the sensor and the controller unit in a single pen-type package, as presented in Figure 4a,b [50]. The device was used to measure TEWL from a wetted artificially created skin and achieved a sensitivity of 0.0068 (%RH/(g/m^2^/h)), meaning that for every gram of water molecules evaporated in time, the device detects tiny changes in the relative humidity of its internal chamber. The significance of Sim et al.’s device is that it also can measure other properties such as skin conductance and hardness, both of which are very relevant to the biomarkers of AD and will be introduced in the following subsections. This multimodality of measurement makes e-health devices more desirable, as they can use several measurands simultaneously to monitor patients more accurately. This idea will be expanded in the conclusion of this section.

#### 2.2.2. Skin Permittivity and Conductivity

As mentioned previously, the severity of AD can be linked with increased water evaporation rates and visible dryness of the skin. Martin et al. concluded that the TEWL and dryness biomarkers are correlated to a decrease in the size of corneocyte cells and a lower concentration of bulk (free) water molecules in the EP below the SC barrier [51]. Therefore, a measurement of water concentration within the SC and the EP is also a valid biomarker for the condition. The presence of free water molecules in the skin is proven to affect composition and therefore change the skin’s physical properties and parameters such as dielectric permittivity and conductivity.

The SC layer can be thought of as a dielectric material, and in the presence of an electromagnetic field, this dielectric is polarized to a certain degree, which is defined by the values of permittivity and conductivity [52]. In circuit theory, the polarization of a dielectric media can be empirically measured by parameters such as impedance, capacitance, and conductance. Several studies have concluded that skin exposed to hydrating conditions is observed to have lower resistance values and higher capacitance values than normal and dry skin [53,54]. Hydrating conditions, such as applying humectant or moisturizing cream, assist the barrier properties of the skin, thereby increasing the ability to retain free water within the SC. The increased water concentration in the SC provides more conductive pathways for ionic charges, thereby increasing the conductivity and the dielectric constant, the latter of which is directly proportional to the value of capacitance and inversely proportional to the resistance [55]. Hence, by recording the values for impedance (capacitance and resistance) or conductance from the skin of AD patients, the severity of the condition can be empirically estimated.

The two predominant non-invasive and painless methods presented in the literature to determine the dielectric response of the skin are conduction measurements and capacitance/impedance measurements. Both require the placement of two or more electrodes on the skin, across which an excitation signal, usually AC voltage or current, is imposed. According to Yokus et al., conduction measurements can be affected by factors on the skin surface such as oiliness, hairiness, and environment temperature [54]. Conduction measurements usually require a small electric current to be passed through the skin, and Abe et al. claim that since hair follicles and sweat glands have weaker permeability barriers than SC, they allow more movement of water molecules and thereby are likely to be a preferred path for current flow [56]. Capacitance measurements require the presence of an electric field rather than electric current, and that is why they are less likely to be obstructed by hair follicles. Moreover, it has been discovered that conductivity measurements suffer from decreased sensitivity when distinguishing lower hydration states [57]. This makes capacitance measurements a better fit for the purpose of determining the severity of AD and distinguishing it from other skin conditions.

The dielectric properties of all materials vary with frequency of the applied excitation signal, and hence the measured impedance and capacitance vary, too. This applies also to all human tissues, including skin, and in 1996, Gabriel et al. formulated a database consisting of the dielectric permittivity and conductivity of various human tissues in the frequency range of 10 Hz to 100 GHz [58]. The database visualized how different tissues have distinct dielectric responses across the frequency spectrum, thereby making it possible to distinguish materials solely on their response. Furthermore, the database included measurements from wet and dry skin, confirming that the properties change with the hydration state of the skin, as the permittivity of wet skin is close to ten times as large as the permittivity of dry skin in the lower frequency range between 100 Hz and 10 kHz [59]. Thereby, this property can be extrapolated to measure differences between severely dry skin such as lesional AD skin and healthy or non-lesional skin.

This measurement technique is called electrical impedance spectroscopy (EIS) and involves sampling the impedance response across a spectrum of frequency bands, generating a unique impedance footprint based on the dielectric properties of the material under test (MUT). Rinaldi et al. used this technique to explore the integrity of the SC barrier of both lesional and non-lesional skin of AD patients over a long period of time [10]. They used a commercial spectrometer called Nevisence^®^ (SciBase AB, Huddinge, Sweden) and were able to characterize both states of skin and revealed a positive correlation between increase in EIS values and decreased pruritus and inflammation on the lesions, indicating that healing can be empirically measured [10,60]. Nicander et al. used the same commercial equipment to detect lipid content within the SC and were successful in discriminating between healthy-looking skin from AD subjects and control skin [18]. The Nevisense^®^ and similar spectrum impedance analyzers comprise a wired handheld probe connected to a bulky processing and visualization unit and are advertised as melanoma detection instruments, but research has proven their applicability toward assaying AD.

Other commercial sensors include a limited or fixed scope of frequencies and thus are sometimes housed in more compact packages, allowing for easier operation and even home monitoring, which is in line with e-health technology guidance. There is not a specified commercial sensor just for assessing AD, but skin hydration sensors have been widely utilized in determining the severity of AD in patients. Gidado et al. present a useful table alongside a comprehensive summary of commercial skin hydration impedance and capacitance sensors that can be used to detect a fault in the skin barrier function, so this review will just highlight the ones that have been tested on subjects with AD [57]. The Corneometer^®^ CM 825 (Courage + Khazaka, Cologne, Germany) is referred to as the “gold-standard” in capacitive skin measurements and is the most widely used equipment in monitoring AD in vivo [61]. The Corneometer^®^ has an interdigitated capacitor (IDC) structure on its sensing electrodes, as seen in Figure 5, which makes the capacitance readout more sensitive to changes in the dielectric permittivity of the MUT. Other notable commercial sensors that employ the same measurement technique are the MoistureMeterSC^®^ by Delfin Technologies and the Novameter^®^ DPM by Nova Technology Corp [62,63].

Grinich et al. compare three different commercial sensors—the Corneometer^®^, the GPSkin^®^ (GPower Inc, Seoul, Republic of Korea) and the TEWL Aquaflux^®^ (Biox Systems, London, UK) to monitor skin barrier function of AD patients. They indicated good correlation (*p* > 0.80) between decreased capacitance values and increased TEWL values from lesional skin, but none of the sensors were able to distinguish between different severity states of the condition [65,66]. Matsumoto et al. employed the Scalar MY-707S Moisture Checker^®^ (Scalar Corp., Tokyo, Japan) to measure the capacitance of the skin of infants, both healthy and with atopic dermatitis, and they measured significantly lower capacitance readouts from atopic skin [63,67]. Sator et al. used the Corneometer^®^ on healthy and AD subjects to conclude that capacitance is not only decreased by the lower hydration values but also because SC lipids have decreased in size. This strengthens the argument that SC hydration is a valid biomarker for AD, because a broken skin barrier results in smaller corneocyte cells (the building block of the SC) [68]. Chiang et al. explored the mean hydration improvement after different treatments applied to atopic skin such as bathing or applying emollient. They used a Novameter^®^ DPM 9003 (Nova, Gloucester, MA, USA) and Scalar Moisture Checker^®^ (Scalar Corp., Tokyo, Japan) and determined that emollient usage assists the skin barrier function, while bathing, although providing an immediate moisturization, dehydrates the atopic skin in the long term [62,69].

While all the commercial sensors have proven to be useful in determining AD symptoms and severity and, due to their relatively small package, allow for mobile/remote monitoring, they still lack the ability to conduct continuous measurements to provide insight into how the condition progresses in time. To achieve a fully non-invasive telemedical device, the skin permittivity sensor needs to be housed in a compact package that can be embedded in a wearable garment or patch to ensure constant contact with the skin. Jang et al. designed a textile-based wearable impedance sensor that was able to accurately distinguish different levels of skin hydration including severely dehydrated skin [70]. The sensor, as shown in Figure 6, comprises two silver electrodes printed onto a flexible cotton substrate, allowing for conformal attachment to the skin. In the low-frequency spectrum, the impedance varies by a factor of 100 between 10 kΩ and 1 MΩ for hydration measurements of moisturized and dry skin, respectively. The textile sensor’s readings were calibrated against hydration readings measured using a commercial sensor [70]. Rie et al. utilized the same IDC design as the Corneometer^®^ but housed the sensor in a device with a total area of 2.3 × 4.6 mm^2^ using CMOS technology. The device was tested for its suitability to measure the dehydration rate of the skin and exhibited higher sensitivity than commercial sensors and can be attached via medical bandage onto the skin [71]. These advances prove that skin permittivity sensors can be embedded into a non-invasive wearable package for the purposes of continuously monitoring AD patients, but so far, no prototype has been presented. Advances in e-health technology and e-textile science can be used to copy the functionality of current commercial skin impedance sensors and package them into a new generation of conformal and wearable sensors.

#### 2.2.3. Skin Elasticity

Skin elasticity is another biomarker of AD that can be empirically measured through telemedical techniques. The increased hydration loss caused by AD makes the skin more inelastic and susceptible to cracking [72]. A single commercial device, called the Cutometer^®^ by Courage + Khazaka GmbH (Cologne, Germany), was used to quantify the elasticity of the skin of AD. The device has a rigid cylindrical hollow probe and a wired control unit and uses a suction method to mechanically deform the skin and a light source to determine how the skin resists the negative applied pressure and its ability to return to the original position [73]. Montero-Vilchez et al. used the Cutometer^®^ to demonstrate that skin elasticity values are lower on AD lesions of patients with the condition than on non-lesional areas or healthy subjects’ skin (0.69 vs. 0.74, *p* = 0.038). They also highlighted that the negative correlation between age and skin elasticity was stronger in AD patients (r = 0.494), meaning that AD increases the inelasticity of skin as the patients age [74]. In another publication, Montero-Vilchez et al. compared the results from the Cutometer^®^ with measurements from a Corneometer^®^ and a TEWL Tewameter^®^ (EnviroDerm, Longhope, UK), concluding that TEWL and SC hydration are better biomarkers for distinguishing between heathy and AD skin because of the bigger difference in the sensor output (TEWL, 9.98 vs. 25.51 g/m^2^/h, *p* < 0.001 and SC hydration 44.36 vs. 24.23 AU, *p* < 0.001, for healthy and AD skin respectively) [75]. Constantin et al. recorded the elasticity values of lesions on patients with dermatitis before and after the application of an emollient cream [76]. They reveal an increase in skin elasticity values after continuous application of the emollient; hence, elasticity measurements can be used not only for distinction but for monitoring the progression of the treatment. Continuous measurements during the treatment process are important because they provide insight into how the skin is responding to the treatment. The results may indicate whether the frequency or dosage of the treatment should be altered. Furthermore, there has not been any investigation of whether skin elasticity can distinguish between different severity levels of AD or between different skin conditions such as psoriasis, urticaria, or melanoma. The lack of a variety of measurement devices (currently, only the Cutometer^®^ is available) can also pose an obstruction to further investigation of the technique.

#### 2.2.4. Scratch Activity and Frequency

A more abstract way of estimating the severity of AD and related conditions is to examine the effects on the daily lives of the patients. Dermatitis is characterized by pruritus and itchy lesions, causing sleep disturbance in the patients, which contributes to decreased performance, daytime fatigue, and irritability [22]. Thus, a record of how frequently the patient scratches their skin would provide insight into the severity of the condition and subsequent treatment needed. Such a measurement can be performed subjectively by the patient self-reporting or objectively with a wearable non-invasive sensor. Wrist actigraphy is an unobstructive technique that uses an accelerometer mounted on a wristband to distinguish between scratching and other nocturnal movements of the hands [22]. Wrist activity measurements are a much better option for monitoring nighttime AD scratches than polysomnography, which is the standard for identifying causes for sleep disturbances using heartrate and respiration, because it is less technically demanding as it tracks just the movements of the wrist [77].

Ebata et al. used a commercial wrist activity sensor called the ActiTrac^®^ (IM Systems, Baltimore, MD, USA) alongside an infrared video camera to map data activity from scratching and noted a significant correlation between the severity state of AD and the activity levels. Wrist activity due to scratching increased to 44 a.u. for AD patients compared to 9 a.u. for controls [21]. Bender et al. used a different commercial device called the MicroMini Motionlogger^®^ (Ambulatory Monitoring, Inc, Ardsley, NY, USA) to conclude that patients with AD have significantly more compromised sleep than controls, as high nocturnal activity measurements are linked with fatigue on the following day [22]. The device is shown in Figure 7d, and the tested subjects have reported no complaints from its continuous usage. Noro et al. present an acoustic evaluation system for detecting scratches in patients with AD, as illustrated in Figure 7a–c. The sensing element is a piezoelectric device which vibrates due to the sound caused by the scratching of the skin, and the post-processing system filters the signal to eliminate noise [78]. Kim et al. used a cloud-based mobile system to source data from textile wrist sensors and analyze it via a web application so that it can be easily accessible to health professionals worldwide [79]. Mahadevan et al. utilized machine learning techniques to sample data from a wristband sensor and to compare it to readings from polysomnography measurements to match sleep states. The wristband showed that scratching episodes match the wakefulness states, when the person is either woken up or sleeps very lightly, thus hindering the patient’s ability to enter deep sleep [77].

Scratch activity sensors are a reliable way of estimating the severity of AD by measuring the disturbance on the sleep cycle of the patient. The device’s low-profile footprint allows for continuous usage throughout the night without stressing the patient. Measurement throughout the day is possible, but currently, a daytime scratch actigraphy sensor has not been demonstrated due to difficulties in distinguishing scratching from other typical daily activity [80]. Further research into these sensors must be provided to explore whether the sensor can also distinguish between other conditions that cause pruritus such as insect bites, infections, or allergic reactions.

#### 2.2.5. Neural Network Imaging Systems

Another telemedical technique that has been used to identify and monitor the lesions of AD in human skin is to employ an artificial intelligence (AI) algorithm, trained using datasets of images or a camera/imaging device, to detect the condition. Hsiao et al. compared the results from identifying mycosis fungoides, AD and psoriasis using a single-shot multibox detector model but obtained low output sensitivity and precision when analyzing AD images (80% and 86%, respectively), concluding that this model has poor performance for AD distinction [81]. Padilla et al. managed to differentiate between AD and Psoriasis using the MobileNet architecture based on a convolutional neural network (CNN). The network was trained using dermatology datasets online and when tested on 30 individuals using a Raspberry Pi camera, it achieved a higher accuracy of 90% when classifying psoriasis and 88% in AD [82]. Pan et al. created an AD-focused computer vision model called the EczemaNet that can identify photographs of the condition and predict the severity and state of the disease [83]. The model achieved a 90% success rate in distinguishing the condition and a low root mean squared error of less than 2 for the severity prediction. An overview of the EczemaNet model is presented in Figure 8. Patella et al. explored the relationship between AD severity and exposure to air pollutants and environmental conditions using an artificial neural network (ANN) [84]. They found out that an increase in the diurnal temperature range (DTR), which is the variation between the highest and lowest temperature in a day, increased the severity of AD lesions by 200%. The ANN can be used to predict disease severity based on the environment and to alert the patients in advance to avoid possible irritants [84].

Neural network systems are a reliable and non-invasive way of classifying eczema and in some cases can effectively estimate the severity of the lesions. They allow for remote monitoring without the need for a clinician or specialist dermatologist, as patients can check their condition through their mobile phone camera. The only drawbacks to ANN are that they cannot be used to monitor the condition continuously, and the models that are currently employed have a maximum reported accuracy of 90%. Further research into this field should seek to increase accuracy and to prototype a mobile device or app that would allow patients to use the power of the algorithm to assay their condition with confidence. Furthermore, experiments should be undertaken to explore the effect of emollient or moisturizing cream on the sensitivity of the ANN to see if treatment progression can be measured.

#### 2.2.6. Conclusions

As presented in this section, there are a variety of telemedical techniques already used to provide empirical measurements toward the monitoring of eczema. These methods all fall in line with the e-health directive for achieving non-invasive and at-home monitoring, but not all of them can be used continuously as a portable device with little impact on the daily routine of patients. The individual advantages and drawbacks of each are summarized in Table 2. TEWL measurements are an accurate way of distinguishing atopic skin and predicting treatment using external relative humidity sensors, but they are occlusive and cannot be used for continuous measurements. Skin permittivity through capacitive and impedance measurements have problems with precision due to disturbances in the skin-sensor contact but can be housed in a non-invasive package for portable and continuous measurements. Furthermore, they have been shown to detect the condition, quantify the severity and monitor the response after treatment. Elasticity measurements have been tested against permittivity and TEWL measurements and demonstrate reduced accuracy in determining AD. The need to provide negative pressure eliminates the possibility of housing the method in a compact wearable package. Scratching actigraphy sensors are a proxy assessment by determining the severity of pruritus severity during nighttime, but these sensors have not been proven to discriminate between different conditions and have limited use during the daytime. Figure 9 illustrates the operation of all non-invasive techniques discussed in this section and reveals their respective measurand. Telemedical instrumentation does not induce any stress or harm onto the patient, and thus, it is the preferred method when compared to invasive medical examinations. Research should be focused on establishing these methods of examination in a clinical setting rather than visual and blood test procedures as they provide an objective measure.

## 3. Future Applicable Technologies

There are other telemedical compatible monitoring methods that have not yet been evaluated for use in AD applications but have been used to quantify similar conditions or to detect some of the biomarkers of eczema. Two methods highlighted here offer the potential for a non-invasive e-health device for monitoring AD, but due to the complexity of the instrumentation used, at present, their footprint is too large and expensive to be embedded into a wearable device and given to patients to use at home. Thus, this review serves as a call for action in developing a more compact prototype sensor based on these methods and exploring that sensor’s response from contact with atopic skin.

Since eczema causes the skin barrier to permeate water molecules at an excessive rate and to dehydrate the epidermal and SC layers, a skin hydration sensor that targets those outermost layers can also estimate AD severity. This sensor would operate on a similar principle as the skin permittivity electrode sensors, i.e., inducing an electromagnetic wave and measuring the reflected response. This electromagnetic wave can be a microwave in the radio frequency (RF) spectrum or an optical light in the near-infrared spectrum. Skin hydration sensors using radiation from these spectra have already been developed, and this section will present the operation of these devices and discuss why they could be viable for distinguishing AD among other conditions.

### 3.1. Radio Frequency Reflectometry

It was previously established that free water molecules (not bound by lipids and proteins) inside the SC and the EP greatly affect the dielectric permittivity of the skin. Water has a significantly higher relative permittivity than dry skin; thus, increased water content would translate into higher permittivity [85]. Using the Cole–Cole expression for complex dielectric permittivity, Arab et al. calculated that at 70 GHz, the relative permittivity of wet skin is increased by 30% and conductivity increases by 10% when compared with dry skin [86]. At the millimeter and microwave frequency range, the effects of relaxation on water molecules dominate the dielectric spectrum of tissues [87,88]. The emitted signals cause excitation of the water molecules, which absorb some of the energy carried by the wave. Thus, by monitoring the intensity of the reflected signal for a given frequency band, the water content can be estimated [89]. Therefore, this technique should be able to measure the dryness state of atopic skin and to distinguish it from similar conditions.

Time-domain reflectometry (TDR) is a popular technique in the literature to quantify the variation in the reflected signal from an MUT. In TDR, the input signal is usually a step-like voltage pulse, which is propagated along the sensing element (SE) or antenna in contact with the MUT with some relative permittivity ε_r_. The TDR method calculates the reflection coefficient 
$\rho \left(t\right)$
, which is the ratio between input and output voltage signals with respect to time, as given by Equation (1) [85]:
(1)
$$\rho \left(t\right)=\frac{V_{refl}(t)}{V_{inc}(t)}$$


When analyzing the reflectogram properties, the relative permittivity of the skin can be obtained from the reflection coefficient. Due to complications in analyzing signals in the time domain, the TDR method can also be transformed into the frequency domain, where the reflection is estimated using the reflection scattering parameter 
$S_{11}$
 [90]. The 
$S_{11}$
 parameter is calculated using a ratio of the discrete Fourier transform (DFT) of the reflection coefficient (Equation (2)):
(2)
$$S_{11}\left(f\right)=\frac{DFT[\rho \left(t\right)]}{DFT[\rho_{i}(t)]}$$

where 
$\rho_{i}(t)$
 is the time-domain reflection coefficient when no material is measured by the SE. The parameter indicates the amount of energy that is lost or absorbed by the MUT. Scattering parameters (S-parameters) are easier to visualize against the measurement frequency and thus allow for better estimation of resonance frequencies of systems under test. The 
$S_{11}\left(f\right)$
 can also be directly monitored using a vector network analyzer (VNA) to further simplify the technique [85]. VNAs are, however, typically bulky, and expensive benchtop devices for visualizing s-parameters, but there are compact models used in the literature that have been integrated into portable monitoring devices. Monti et al. used a nano VNA (HCXQS, Nanjing, China) operating in the frequency range of 0 to 3 GHz, which is sufficient for RF reflectometry, but this is still quite bulky and has a high-power consumption [91]. The output of frequency-coupled TDR, as visualized by a VNA, is a plot of the 
$S_{11}$
 parameter against the frequency. Schiavoni et al. used a parallel electrode SE in contact with three different states of skin based on their hydration. Setup and produced TDR readings are shown in Figure 10a,b.

It can be seen from the plot that as the hydration levels increase, the magnitude of the reflection parameter around the resonance frequency of the skin-sensor system increases [90]. This is due to the increased permittivity and conductivity in hydrated skin, and this demonstrates the ability of TDR within the microwave frequency range to detect hydration changes in the skin.

Table 3 presents a summary of the research papers that have investigated RF reflectometry for the purposes of measuring dry hydration state of the skin. Monti et al. present a tri-electrode sensor that can distinguish between three different states of hydration in the skin [91]. They discovered that around the resonance frequency of the system, the reflection parameters of dry skin are significantly lower, which means that dry skin lesions can be discriminated from healthy non-atopic skin. Schiavoni et al. utilized the tri-electrode sensor and integrated it in a wearable and flexible strap for mobile measurements, and Cataldo et al. extrapolated its functionality to predict the hydration state of extra dehydrated skin [85,90]. Eczema is characterized with severely dry skin, and Cataldo et al.’s sensor should be able to distinguish this from normal skin and produce a reflection parameter value of around 0.165 based on the dry state prediction. Arab et al. fabricated a sensor with an open-ended coaxial probe that performed millimeter-wave reflectometry on skin cancer (melanoma) lesions [86]. Melanoma skin has distinct dielectric properties, much like atopic skin, and Arab et al.’s device demonstrated that the reflection parameters from melanoma skin are different to those of healthy dry or wet skin with an accuracy of tens of microns in lesion size [86]. While their design is not suitable for mobile measurements, it proves the viability of RF reflectometry for detecting skin conditions. Mehta et al. employed a smaller coaxial probe to differentiate between normal skin, benign and malignant melanoma [92]. This is a very important distinction because it means that RF reflectometry might be utilized to discover skin conditions before the visual symptoms appear. Gao et al. use the reflectometry technique to classify burn degrees and can successfully distinguish three different degrees of burnt skin [88].

RF reflectometry has been demonstrated as an accurate tool to measure severely damaged skin and could be used to monitor the most severe cases in AD and psoriasis. Further research is required to explore the response of atopic and eczema skin under RF reflectometry and to see if it could be distinguished from other skin conditions such as melanoma and psoriasis. In addition, smaller, lower cost and low-power alternatives to conventional VNAs are required for wearable, home based application of the approach.

### 3.2. Optical Spectroscopy

When light from the optical spectrum (visible light and infrared with wavelengths from 400 nm to 1 mm and frequencies from 800 THz to 300 GHz) interacts with tissue or media, the phenomena of light scattering and light absorption occur [94]. During the process of scattering, the photons traverse in various directions among the cells or blood vessels until they are absorbed or remitted back outside of the tissue. The number of photons that become absorbed or reflected is a probability function that depends on the inherent energy of each photon. This function can be quantified using the intrinsic optical properties of the tissue: the absorption coefficient (µ_a_) and the scattering coefficient (µ_s_) [94]. These properties depend on the chemical composition of the tissue and especially on the concentration of water molecules in the case of the human skin. Therefore, measurement of the optical absorption by analyzing the input and output signals incident on the skin’s surface can provide insight into the water composition within the skin.

To calculate the light absorption (*A*) through a specific media, the Beer–Lambert law is used (Equation (3)).

(3)
$$A=\log_{10}\frac{I_{0}}{I}=\log\frac{1}{R}$$


Here, 
$I_{0}$
 is the intensity of the input light, 
I
 is the intensity of the output, and *R* is the reflectance [95]. The output intensity depends on the absorption and scattering parameters, µ_a_ and µ_s_, and on the distance 
$\rho_{0}$
 between where the incident light enters the tissue and where the reflected light exits. The ratio between the input and output is the reflectance, and from it, the absorbance can be calculated using Beer’s law. Modern spectrometers perform this calculation automatically and provide absorption measurements directly in units of log (1/*R*). Since the µ_a_ and µ_s_ coefficients depend on the frequency/wavelength of the incident light, the absorption values are commonly graphed on a wavelength spectrum. Absorbance peaks at specific wavelengths because some combinations of molecules begin vibrating and absorbing photons at an accelerated pace [13]. The same is valid for other measurands arising from Beer–Lambert’s law such as the reflectance or intensity of scattered signal, which can be obtained through various measurement techniques and plotted against the frequency. This frequency-dependent behavior provides insight into the chemical composition of the MUT by highlighting distinguishable segments of absorption/reflectance increase, which indicates the presence of specific compounds [95].

There are various methods of performing these measurements based on the frequency range, equipment setup, light source and the targeted measurand. This review covers methods demonstrated in the literature as viable skin hydration sensors and in some cases that have been used on skins with eczema. Raman spectroscopy focuses on the scattering of light, and it measures the intensity of scattering events occurring at specific wavelengths, thereby estimating the concentration of molecules with known scattering effects at these frequencies [96]. Near-Infrared Spectroscopy (NIRS) monitors the absorption intensity from the reflectance spectrum of an exposed tissue or material within the near-infrared band (wavelengths of 0.8 to 2.5 mm). The absorption is linearly correlated with the concentration of specific molecules at unique band levels, and by selecting the specific NIR waveband, insight into the presence of individual molecules can be determined [97]. Diffuse reflectance spectroscopy (DRS) is another widely used technique to determine chemical composition by measuring the remitted light after interaction with the MUT. It considers the lateral distance traveled by the light within the MUT, affected by scattering and absorption events, allowing for a spatially resolved measurement [98]. The methods described here are all accurate enough to be used as AD monitoring devices, but a fully portable and mobile version of these devices has not been developed yet.

#### 3.2.1. Raman Spectroscopy Measurements on Skin

Confocal Raman Micro-Spectroscopy (CRM) is a type of Raman spectroscopy widely used to perform non-invasive in vivo measurements on the SC and EP layers of the skin, as it can determine the biomolecular composition at depths up to several hundred micrometers [96]. The method uses the variance in scattered protons depending on the chemical content to provide a unique signature Raman spectrum for each material measured. AD affects the chemical composition of the skin not only by reducing the free water molecules inside but also eliciting other molecular biomarkers. Dev et al. explored the Raman intensity at different wavebands from 400 to 1800 cm^−1^ to classify the differences between atopic and healthy skin [96]. The probe used was handheld and portable, but the light source and the spectrograph required for the measurements are bench top equipment, and therefore, this approach is not portable, as visualized in Figure 11. They record differences in the Raman spectra at the wavebands corresponding to water, ceramide and urocanic acid, the latter of which features the highest separation at the wavelength of 6098 nm [96]. This is an important metric for the discovery of other biomarkers of AD. Ho et al. used the same method to discover that eczema skin had 51% and 52% lower concentration of water and urocanic acid, respectively, compared to healthy control skin [99]. Their device was also tested against different severity levels and was able to distinguish between mild and moderate but not between moderate and severe states of eczema. Results were also compared with TEWL measurements, and the TEWL exhibited a 32% increase in sensitivity compared with CRM in distinguishing between the mild and moderate severity states [99]. Gonzalez et al. used a commercial benchtop Raman spectrometer to detect AD early in infants based on the presence of a key skin barrier maintenance protein called filaggrin (FLG) [100]. It was discovered that the infants with lower FLG concentration were the ones that developed AD, identifying it as a valid biomarker for the condition.

CRM is a highly accurate tool for discriminating the biomarkers of AD, but its requirement of a multi-lens arrangement, along with a halogen light source and CCD spectrograph make it unsuitable for portable, home-based application at present.

#### 3.2.2. Near-Infrared Spectroscopy Measurements on Skin

The absorbance spectrum of water and porcine skin in the near-infrared range (NIR) is compared in Figure 12. It is evident that the skin spectrum follows the spectrum of the water and peaks at the same wavelengths of around 1450 and 1900 nm. This is because at these bands, the combinations of OH and HOH molecule groups of water begin vibrating [13]. These groups are attributed to the bulk water content within the SC and the EP, which is one of the biomarkers of AD. Therefore, skin with lower free water content, such as atopic skin, will have lower absorbance values at these specific wavelengths than normal, healthy skin and a device that can estimate the scattering, absorbance, or reflectance of light through the skin would be useful in monitoring skin conditions such as AD.

NIR Spectroscopy (NIRS) devices perform absorbance measurements across the frequency range to detect deviations between the measurand and a control unit, indicating the differences in the chemical composition between the two. NIRS has already been employed and widely used in determining melanoma spots, as Fioravanti et al. have presented a portable, but not yet wearable, device that complements histopathological measurements to prevent misdiagnosis [101]. The wavelength used is tailored to detect discrepancies in the methylene absorption, as it is an indicator for melanoma. By changing the frequency, it should be possible to detect the biomarkers of AD such as water molecules, FLG, ceramide and chemokine [16,51,100,102]. Shin et al. have used NIR to monitor molecule changes during acute barrier disruption in the SC, which was artificially induced via tape stripping. They have discovered the same peaks of methylene absorption as with melanoma as well as additional insight into the protein and lipid composition of the SC [103]. This confirms that NIRS can analyze the lamellar structure of SC and monitor the ceramide protein levels, rendering it another possible approach for monitoring AD. Zhang and Meyers et al. have created an NIR imaging system using an InGaAs array detector and two tungsten halogen lamps to detect hydration changes of the skin in the range 800–1800 nm [97]. The skin was treated with different products such as humectant cream and moisturizing body wash, which are both remedies for the symptoms of AD. The results were compared to visual and electrical methods of hydration assessment, and it was shown that the NIR method related more closely to the visual assessment performed by a clinician than to the electrical ones, even though the latter showed higher sensitivity to the humectant cream [97].

NIRS has the potential to be an accurate tool of identifying the biomarkers of AD but has not yet been tested to assay the condition, but Zhang and Meyers et al. stated the method cannot target the SC and EP layers for hydration detection and that it measures the overall hydration in the skin [97]. Further research is required to determine if NIRS can distinguish AD from other skin conditions such as skin burns or that the dehydrated state of just the SC cannot be measured.

#### 3.2.3. Diffuse Reflectance Spectroscopy Measurements on Skin

Diffuse reflectance spectroscopy (DRS) is commonly performed in the IR range, and it utilizes a two-probe arrangement, connected with a spectrometer, to establish the optical parameters of an MUT. The optical probes are employed as an emitter and collector and are positioned on the surface of the MUT with a defined separation between each probe [94]. When an emitted photon package passes through the MUT, part of it is absorbed, and the rest undergoes scattering and emerges at some distance from the input. The levels of absorption and scattering will vary depending upon the inherent optical properties and coefficients of the MUT [95]. From the intensity of the reflected signal and the distance it has diffused from the input, a depth-resolved image of the MUT can be estimated, thus providing insight into the chemical composition of tissues at different layers. The separation distance between the source and detector is determined by the probe placement, and by varying this distance, measurement at specific depths can be achieved [95]. Therefore, the DRS sensor can be set to detect the concentration of AD biomarkers within the SC and the EP at a specific point on the skin using the two probes and electronics for analyzing the reflected light.

Since DRS is operating in the IR range, and the absorbance peak bands of specific molecules in the IR range are already known, the total footprint of a DRS measurement device can be reduced to a simple light-emitting diode (LED) with a fixed wavelength and a photodiode (PD) or camera to capture the reflected signal. The wavelength of the LED will be matched to the absorbance peaks of the relevant AD biomarker molecules such as water, methylene, FLG, and ceramide. Anker et al. have used a multi-LED arrangement around a camera setup to visualize the epidermal keratin and dermal collagen structure for the purposes of understanding abnormal keratin formation in ichthyosis, which is a condition similar in severity to AD and commonly in infants [104]. They were able to link the abnormal keratin structure to gene mutations, such as the mutations in AD patients that cause the breakdown in the linkage proteins within the SC barrier. To optimize the DRS sensor arrangement, multi-wavelength LED setups can be employed to detect other absorption peaks. Yan et al. have presented an opto-electronic patch sensor that features four channels of different wavelengths ranging from 525 to 870 nm around a single PD cell [105]. The sensor (visible in Figure 13) is housed in a compact and wearable packaging and was used to detect the heart rate of the subject while performing daily activities.

Mamouei et al. employed the same multi-LED sensor arrangement but used three LEDs with wavelengths of 970, 1200 and 1450 nm to determine dermal skin hydration. These wavelengths correspond to the absorbance peaks in Figure 13 [13]. The sensor has been simulated to target the EP and dermis layers, but by tweaking the SDS distance, the penetration depth can be altered to target the AD-relevant SC layer. The device is incorporated into a coin-sized non-invasive package and has been tested on skin phantoms using gravimetric testing, which involves wetting the skin phantom and positioning it onto a fine scale and recording the weight of the evaporated water from the surface of the phantom [13]. The authors claim that the sensor has noise immunity from other physiological influences on the skin, but it has not been tested in vivo, so further testing is required.

The standard two-probe DRS method is limited to monitoring a single point, but with an image filter or optic plate, the measurement can be extended over a two-dimensional area. This type of DRS is called diffuse reflectance imaging (DRI) [94]. DRI can be performed over an area of pixelated sensor probes to gather spatially resolved information about the MUT [98]. The source of light within DRI would be a single LED pixel, which is usually placed in the middle of a matrix of PD pixels to pick up the reflected signal. Such an arrangement would not only allow water content estimation across an area but would be able to pinpoint how the water content varies with depth. If the water molecule absorption at the shallower layers is low, then photons will traverse deeper into the tissue and would remit further from the source [94]. Thereby, since AD skin is dryer, the intensity measurement of the outer pixels should be higher than that of the inner ones. Petitdidier et al. proposed such a spatially resolved DRI sensor using CMOS technology to fabricate a pixelated sensor with a single light emitter and a matrix array of PDs to determine the absorption and scattering coefficients of the skin [98]. A fiber optic plate (FOP), which is a grid of parallel vertically oriented fibers, guides the reflected light from the surface of the MUT to the pixels of the sensor. The sensor was able to accurately calculate the optical properties of the skin but has not been tested in vivo on skin conditions to determine its efficacy.

#### 3.2.4. Conclusions

Optical spectroscopy is a valid future method for determining the physiological characteristics of damaged skin and has already proved its usability in monitoring conditions similar to AD. Optical-based approaches are immune to electrical noise arising from hair follicles and skin pores, and their response is not affected by the skin–sensor contact pressure, unlike methods such as RF reflectometry and electrical impedance. Table 4 summarizes the most prominent spectroscopic devices. The methods require further work to evidence their usability for examining AD specifically. Raman spectroscopy is the only optical method that has been tested on AD patients, but due to the complexity of the equipment, it cannot be classified as a portable e-health device that does not require the input of an expert operator. Further research to reduce the size of spectroscopic devices and simplify their implementation is required to enable them to be used by the patient in a home monitoring scenario.

## 4. Discussion and Conclusions

Advancements in e-health technologies and devices have paved the way for patients to receive medical treatment and expertise from the comfort of their home. Digital and telemedical health services have become increasingly prevalent since the global outbreak of the COVID-19 pandemic and have highlighted the need for more research to further expand capability. Mbunge et al. comment that since the pandemic, there has been a boom in the development of sensors embedded into smartphones, smart wearables and IoT devices, as an alternative way of sourcing health data and monitoring patients when attending hospitals is not possible [106]. Dwivedi et al. conclude that wearable sensors would also transform conventional disease diagnosis by utilizing data analysis, machine learning, and artificial intelligence instead of relying on a doctor or clinic to perform the diagnosis, access to which has been very limited ever since the COVID-19 pandemic [107]. Diagnosing and treating skin conditions such as AD is still typically performed within a clinical environment which is restrictive, time consuming and costly. AD monitoring is also typically performed by specialists often using often visual assessment rather than invasive examination methods such as biopsy, tape stripping and blood testing. This introduces a level of subjectivity when determining the severity of the condition, which can affect the proposed treatment.

This review has explored a wide range of possible techniques, both in the prototyping and conceptual stage, that could provide a solution to AD monitoring in the home. This study is not confining to AD only, but in some cases, it also applies to other conditions such as psoriasis, ichthyosis, skin burns and melanoma. The focus on home-based techniques required an assessment of the level of invasiveness, evaluation of the platform (e.g., wearable), ease of use, accuracy, the cost and complexity of the approach and the suitability for longitudinal monitoring.

TEWL measurements are accurate enough to monitor AD but are occlusive and thereby cannot be used for continuous monitoring. Electrical-based impedance measurements that quantify the skin permittivity can distinguish between similar conditions and can avoid occlusion but incur problems with contact impedance, noise and pressure issues. Elasticity measurements have been used to monitor AD but are less accurate than TEWL and skin permittivity methods and require an external pump to provide negative pressure for the skin stretching. Wearable sensors that detect scratching assist in determining the severity of pruritus without obstructing the daily routine of patients, but these sensors cannot, on their own, discriminate between AD and other conditions. Neural network systems have been used alongside a camera or with the aforementioned sensors as a separate layer assisting in the classification of the condition and can predict possible complications or assign suitable treatments.

Of the possible mentioned techniques, the electric-based ones are the most prominent because they have been proven to fulfill all criteria of an e-health device being compact, non-invasive, and easily accessible to patients worldwide, and they have shown sufficient capability in predicting the symptoms, evaluating the severity, and monitoring the treatment of AD. Furthermore, the electric-based sensors are the cheapest to produce in terms of material and component costs, and they do not require complex post-processing devices, such as a vacuum pump in the case of elasticity measurements or a sophisticated camera setup.

New technologies such as RF reflectometry and optical spectroscopy may be able to overcome the limitations of impedance permittivity sensors but have not yet been demonstrated in a compact and non-invasive e-health AD monitoring devices. RF reflectometry sensors offer higher measurand sensitivity per sensor area than impedance ones and with reduced sensitivity to motion artefacts but include bulky electronics that largely prevent their use as stand-alone devices [85]. The sensing element is not much different in terms of production costs to that of the electric-based approach, but the processing electronics, such as the nanoVNA for example, which costs around 300 USD (Nanjing, HCXQS), increases the total sensor price substantially. Optical spectroscopy methods are less susceptible to interference arising from electrical noise and contact pressure than electrical-based methods. Measurements in the optical spectrum are more accurate than those in the RF and electric signal spectrum because techniques such as Raman spectroscopy, NIR and DRS can visualize and distinguish between spectra of different molecules, as each type of molecule has a unique absorbance spectrum at given wavelengths. Hence, DRS is better able to detect specific types of AD biomarker molecules within the SC and EP than capacitive measurements. The versatility of the system to monitor other skin conditions using the same setup is also a benefit. The extra accuracy and sensitivity come at the cost of using expensive and often bulky equipment—standard CCD cameras and laser sources can cost much more than any of the other techniques mentioned in this paper. The current power and operational requirements of such optical systems typically render it unusable for integration into wearable electronics [54]. The response of the optical setup can also be heavily influenced by deformation or movement, which is present in wearable systems, and the suitability of optical systems for continuous monitoring has not been demonstrated [108]. New and compact devices such as the CMOS IC and the LED-PD sensors may be able to overcome those limits, but current reports of such sensors are limited.

It is also important to note that these techniques can be combined simultaneously in a multi-modal sensing device to yield the best results. Sim et al.’s TEWL probe also measures the hardness and conductivity of the skin by using a pressure sensor and a simplified version of an impedance sensor’s circuitry [50]. Rie et al. has used CMOS technology not only to miniaturize the IDC impedance sensor but also to be able to embed PD pixels for UV light detection [71]. This proves that potentially an optical and impedance screening device can be developed to arrive at a high-performance single sensing solution for monitoring and diagnosing AD on patients in vivo.

This review has provided a comprehensive summary and a critical analysis of the current literature in the field of empirical sensors for AD. The medical examinations were introduced, along with their drawbacks, in order to contrast them to the new generation of digital, non-invasive devices that have shown reliability and accuracy in diagnosing AD in patients. Furthermore, different techniques such as RF reflectometry and optical spectroscopy are presented, which are relevant to the field, but have not yet seen application in it. Thus, this review invokes action to conduct research work into optimizing the current electrical sensing solutions and into prototyping an RF or optical sensor into a wearable, non-invasive package.

## Figures and Tables

**Figure 1 sensors-23-03935-f001:**
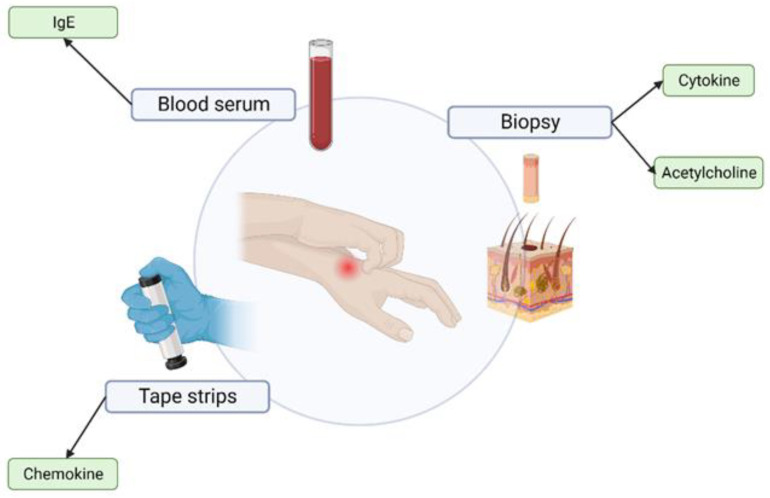
Methods of medical examination of AD and their corresponding potential biomarkers. The three main empirical methods are listed, all of which are performed under a controlled clinical environment. Figure created with BioRender.

**Figure 2 sensors-23-03935-f002:**
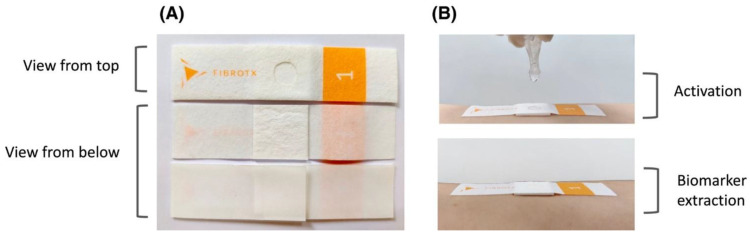
Tape-stripping procedure illustrated with pictures. (**A**) presents the top and bottom view of the strip and (**B**) demonstrates the activation process used to extract the biomarkers. Image reproduced with permissions from [33].

**Figure 3 sensors-23-03935-f003:**
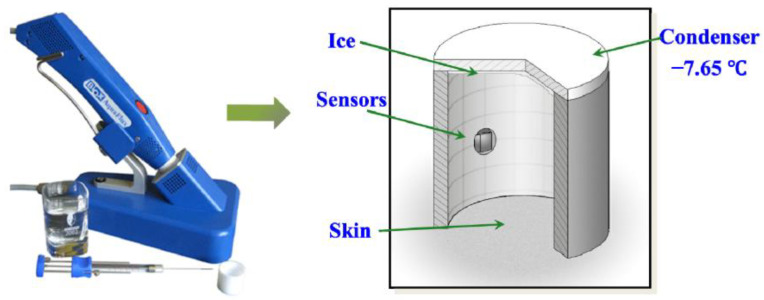
AquaFlux evaporimeter and its measurement principle. The chamber visible on the right is housed within the tip of the AquaFlux evaporimeter and contains the humidity sensors that detect the TEWL. Image reproduced with permissions from [47].

**Figure 4 sensors-23-03935-f004:**
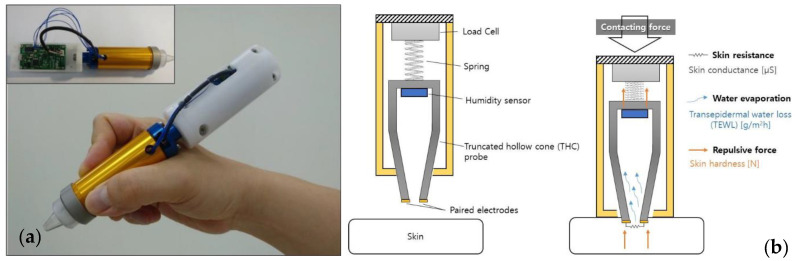
Images of multimodal device for measurement of TEWL, skin conductance and skin hardness constructed by Sim et al. A photograph is visible in (**a**) (left) and in (**b**) (right), its operating principle is presented. Images reproduced with permission from [50].

**Figure 5 sensors-23-03935-f005:**
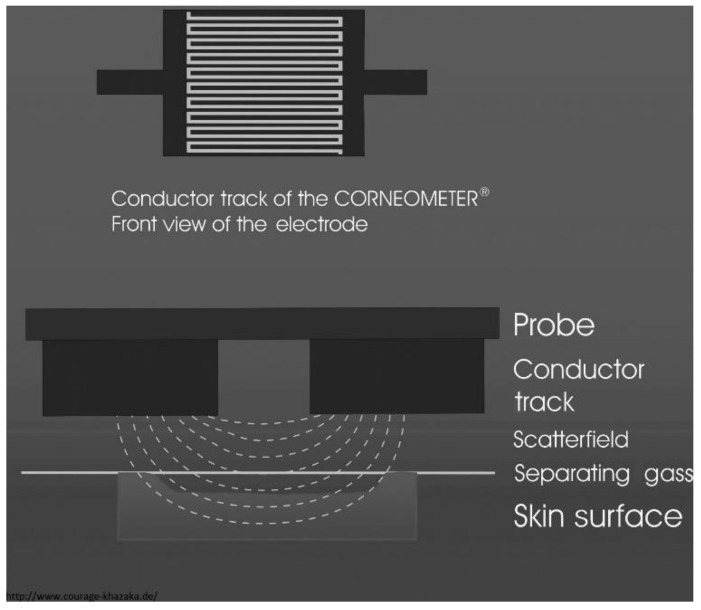
Measurement principle of the Corneometer^®^ CM 825. On the top is the front view of the electrode, revealing its interdigitated shape and on the bottom is the side view, illustrating the scattered electric field. Image reproduced with permission from [64].

**Figure 6 sensors-23-03935-f006:**
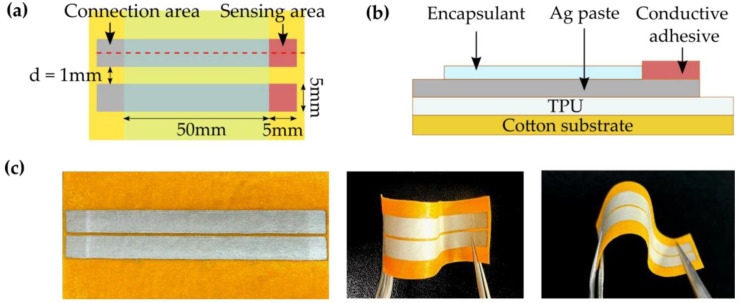
Images of the textile-based impedance sensor created by Jang et al. (**a**,**b**) present a schematic on the top (**a**) and side (**b**) view of the sensor, respectively, and (**c**) demonstrates its flexibility through photographs. Images reproduced with permission from [70].

**Figure 7 sensors-23-03935-f007:**
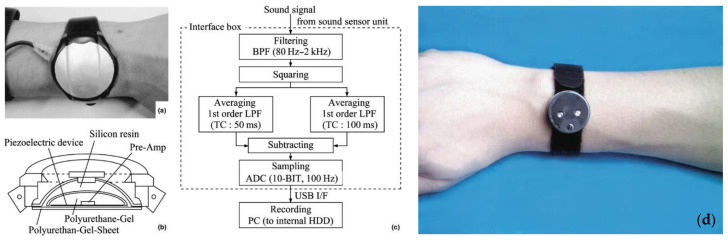
Examples of actigraphy sensors for scratch detection. (**a**–**c**)—Photograph, schematic and flow chart of the acoustic scratch detection sensor by Noro et al.; (**d**) photograph of the MicroMini Motionlogger^®^ to showcase its compactness and non-invasiveness. Images reproduced with permissions from [22,78].

**Figure 8 sensors-23-03935-f008:**
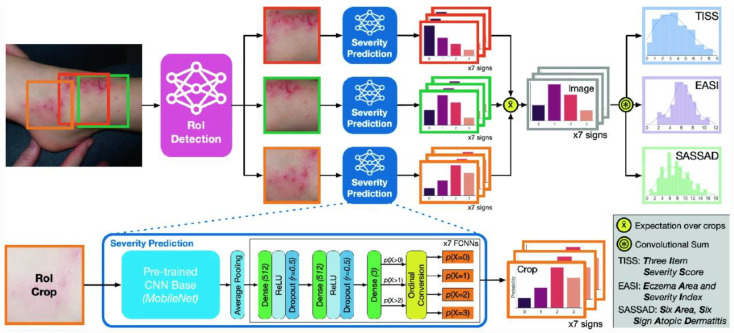
Schematic of the EczemaNet model created by Pan et al. The model can identify photographs of AD and predict the severity of the disease. Image reproduced with permissions from [83].

**Figure 9 sensors-23-03935-f009:**
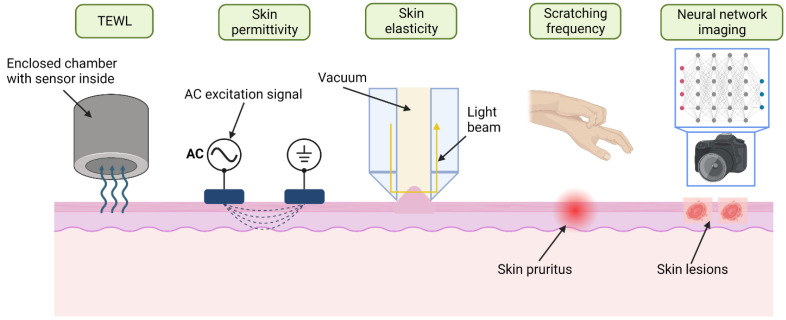
Illustration of different non-invasive telemedical methods for monitoring AD in vivo. The respective measurand parameters are presented on the top of each method’s illustration. The TEWL method features an enclosed chamber with a relative humidity sensor inside so that water vapors from the skin affect its reading. The skin permittivity sensor consists of 2 or more electrodes in contact with the skin of which one is a signal source and the other one is a signal pathway (ground) to establish a scattered electric field through the skin. The skin elasticity sensor uses negative pressure to suction parts of the skin and emit light through it to test the skin’s properties. The scratching frequency sensor detects when a person is scratching their AD lesions. The neural network imaging system uses a camera to distinguish lesions of AD from other similar conditions such as melanoma, skin burn, and psoriasis.

**Figure 10 sensors-23-03935-f010:**
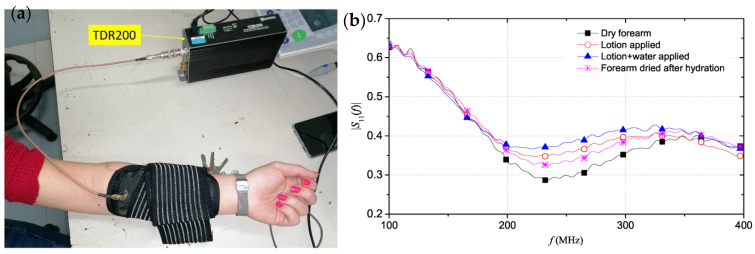
Experimental setup and results of RF reflectometry measurements performed by Schiavoni et al. (**a**) Presents a photograph of the setup with the VNA used to perform the TDR calculations and (**b**) measurement of s-parameters for four cases of forearm hydration. Images reproduced with permission from [90].

**Figure 11 sensors-23-03935-f011:**
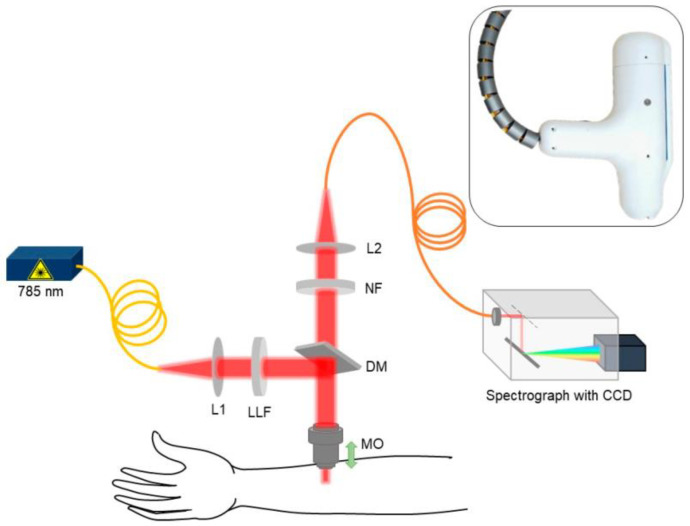
Schematic of handheld in vivo CRM system utilized by Dev et al. The system is housed within the handheld probe visible in the top right corner—the cable on the left is connected to the laser source and the cable connecting to the spectrograph is not illustrated. Image reproduced with permissions from [96].

**Figure 12 sensors-23-03935-f012:**
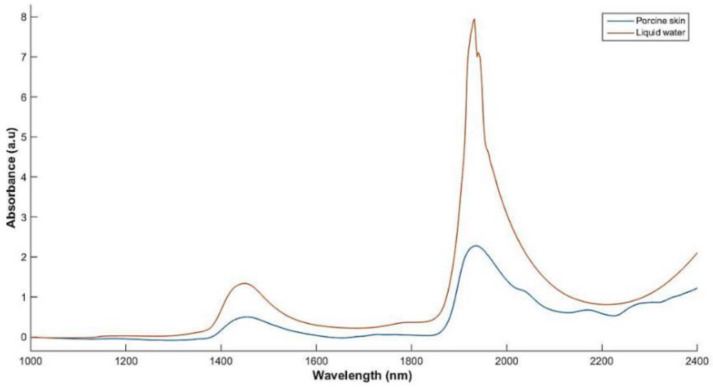
Absorbance spectrum of water and porcine skin in the NIR range. The absorbance peaks around 1400 nm and 1900 nm match almost perfectly, indicating that the absorbance response in the skin is due to the water concentration. Image reproduced with permissions from [13].

**Figure 13 sensors-23-03935-f013:**
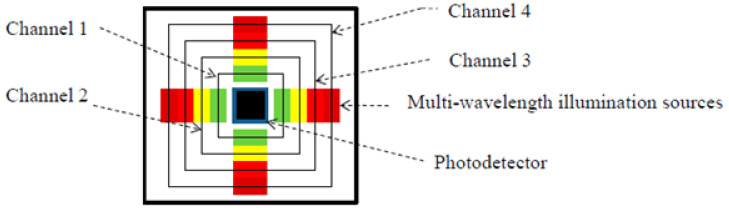
Schematic of opto-electronic patch sensor created by Yan et al. Four channels of NIR LEDs are positioned around a single PD cell. The channel can emit light with varying wavelengths to achieve a higher sensitivity. Image reproduced with permissions from [105].

**Table 1 sensors-23-03935-t001:** Empirical medical tests for diagnosing and monitoring the severity of AD in vitro. None of the methods can be applied in a remote (non-laboratory) location, and all require a specialist operator.

Method	Monitor and Distinguish?	Estimate Severity	Can Predict Condition?	Biomarker	Invasive?
Biopsy [26,27,28,29]	Yes	Yes	Yes	IgE	Yes
Blood serum [11,28,30,31]	Yes	No	Yes	Cytokine Acetylcholine	Yes
Tape stripping [32,33,34]	Yes	Yes	No	Chemokine Ceramide	Yes

**Table 2 sensors-23-03935-t002:** Summary of telemedical solutions used in the literature for monitoring and distinguishing AD in patients.

Method	Monitor and Distinguish?	Estimate Severity	Monitor Treatment	Allows Continuous Monitoring	Wearable or Portable?	Occlusive
TEWL [43,44,45,46,47,48,49,50]	Yes	Yes	Yes	No	Yes	Yes
Skin permittivity [10,18,61,62,63,64,65,66,67,68,69,70]	Yes	Yes	Yes	Yes	Yes	Yes
Skin elasticity [72,73,74,75,76]	No	Yes	Yes	No	No	N/A
Scratching frequency [22,77,78,79,80]	No	Yes	Yes	During nighttime	Yes	No
Neural network [81,82,83,84]	Yes	Yes	Yes	No	Can be	No

**Table 3 sensors-23-03935-t003:** Summary of literature findings of RF reflectometry for the purposes of AD and skin condition monitoring.

Authors	Probe Type	Measurement Frequency	Sensor Placement	Range in S11 Values between Dry and Wet Skin States	Remarks
Cataldo, A. et al., 2022 [85]	Tri-electrode array	0–600 MHz	Forearm and leg	0.165–0.268	Predicts values for extra-dehydrated skin.
Schiavoni, R. et al., 2021 [93]	Tri-electrode array	100–400 MHz	Forearm	0.185–0.290	Explores three skin hydration states. Housed in a wearable and flexible patch
Arab, H. et al., 2020 [86]	Open-ended coaxial	77 GHz	Hand	0.2–0.26	Tested on melanoma skin.
Brendtke, R. et al., 2016 [89]	Open-ended coaxial	7.35 GHz	Tested on phantom	-	Tested on various saline solutions in contact with skin phantom.
Monti, G. et al., 2021 [92]	Tri-electrode array	20–2000 MHz	Forearm, leg, arm, palm	0.224–0.282	Uses a compact VNA for mobile measurements.
Mehta, P. et al., 2006 [92]	Open-ended coaxial	300 MHz–3 GHz	Forearm, cheek, palm, chest	0.08–0.14	Used for melanoma detection. Differentiates malignant and benign lesions.
Gao, Y. et al., 2017 [88]	Open-ended coaxial	26.5–75 GHz	Tested on pig skin	-	Classifies burn degrees.

**Table 4 sensors-23-03935-t004:** Summary of literature findings of optical spectroscopy for the purposes of AD and skin condition monitoring.

Authors	Technique	Sensor Placement	Sensor Footprint	Skin Condition	Wavelength	Remarks
Dev et al., 2022 [96]	Confocal Raman Micro-Spectroscopy	Volar forearm	Handheld probe, connected to bench spectrometer	AD	5550–25,000	Highest sensitivity to urocanic acid.
Ho et al., 2020 [99]	Confocal Raman Micro-Spectroscopy	Volar forearm	Handheld probe with built in spectrometer and light source; bulky package	AD	5000–16,000	Compares device to TEWL and conventional AD scoring system.
Gonzalez et al., 2011 [100]	Raman Spectroscopy	Right thigh	Commercial benchtop Raman spectrometer	AD	5550–50,000	Tested on infants.
Fioravanti et al., 2016 [101]	Near-Infrared Spectroscopy	Biopsy	Multi-lens benchtop setup, connected to a computer and IR emitter	Melanoma	3350–3550	Detects abnormalities of methylene absorption in melanoma patients.
Shin et al., 2018 [103]	Near-Infrared Spectroscopy	Facial cheek	Commercial benchtop spectrometer	Barrier disruption	1450–2500	Performs skin stripping to induce barrier disruption. Can monitor protein and lipid alterations.
Zhang et al., 2005 [97]	Near-Infrared Spectroscopy	Outer lower leg	Benchtop detector and camera setup with two separate light sources	Skin hydration	1100–1630	Claims NIR cannot be used to target individual layers. Compares against electrical methods.
Anker et al., 2021 [104]	Diffuse Reflectance Spectroscopy	Various locations on infants’ skin	LEDs around a camera.	Ichthyosis	405	Monitors mutations of keratin-forming genes.
Yan et al., 2017 [105]	Diffuse Reflectance Spectroscopy	Palm	Multi-LED and PD patch of size 324 mm^2^	Physiological skin changes	525–870	Compact and wearable package.
Mamouei et al., 2021 [13]	Diffuse Reflectance Spectroscopy	Skin phantom	Coin-sized Multi-LED and PD probe	Skin hydration	970–1450	Has not been used on skin, only on fake tissue to predict water evaporation.
Petitdidier et al., 2021 [98]	Diffuse Reflectance Spectroscopy	Skin phantom	Coin-sized pixelated sensor with holder	Physiological skin changes	645	Uses CMOS technology for very small separation distances.

## Data Availability

Not applicable.

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
