# Peer review of "Electromagnetic Sensing Techniques for Monitoring Atopic Dermatitis—Current Practices and Possible Advancements: A Review"

_sensors, 2023, doi:10.3390/s23083935_

Round 1
Reviewer 1 Report
Todorov et al. have provided a comprehensive review on biosensing technologies for Atopic Dermatitis (AD) monitoring. They demonstrated the significance and established methods for monitoring AD. However, the provided figures should be substantially improved.
(1) The first figure should cover the mechanism, methods, and other classifications. The authors should provide an original figure rather than using others’ figures.
(2) The authors are suggested to provide a summary of the major methods listed in Section 2.
(3) Please provide a topic sentence in each figure caption.
Minor comments:
Please cite more recent publications in the references.
Reviewer 2 Report
This paper reviewed an atopic dermatitis very carefully. The current technologies are all covered. It is a solid journal paper and I have only a few comments.
Title of 2.2.4 Quality of Life Measurements is not clear. The daily activities are referred to as AD evaluation..
Line 208- To unify the product name,type, manufacturers, and location.
Also please check the product name in Lines 320 and 331
In table1, it is convenient to show the theory with commercial devices and references.
Line 563 Please provide the location of HCXQS,
Line 806 Missing chapter 4.
Line 810 Discussion; If the authors find digital health, telemedicine and telecare services including Covid-19 please comment on them.
The cost of devices needs to be discussed. The optical spectroscopy is rather expensive compared with wearable devices.
Please add the future prospect as a summary.
The reference should follow the ACS style. The authors need to correct the order of the name (family name is a first).
Round 2
Reviewer 1 Report
The current version can be accepted for publication.